# Study on the Mechanical Properties of Continuous Composite Beams under Coupled Slip and Creep

**DOI:** 10.3390/ma16134741

**Published:** 2023-06-30

**Authors:** Hongliang Nan, Peng Wang, Qinmin Zhang, Dayao Meng, Qinan Lei

**Affiliations:** Faculty of Civil Engineering and Mechanics, Kunming University of Science and Technology, Kunming 650500, China; hongliangnan@stu.kust.edu.cn (H.N.); kustzqm@163.com (Q.Z.); mengdayao100@163.com (D.M.); leiqinanletter@163.com (Q.L.)

**Keywords:** steel–concrete continuous composite beam, variational methods, axial force, deflection, interfacial slip, creep, ANSYS

## Abstract

Steel–concrete continuous composite beams are widely used in buildings and bridges and have many economic benefits. Slip has always existed in composite beams and will reduce the stiffness of composite beams. The effect of creep under a long-term load will also be harmful. Many scholars ignore the combined effects of slip and creep. In order to more accurately study the mechanical properties of steel–concrete continuous composite beams under long-term loads, this paper will consider the combined actions of slip and creep. By combining the elastic theory and the age-adjusted effective modulus method, the differential equation of the composite beam is derived via the energy variational method. The analytical solutions of axial force, deflection and slip under a uniform load are obtained by substituting the relevant boundary conditions. The creep equation is used to simulate the behavior of concrete with time in ANSYS. The analytical solution is verified by establishing a finite element model of continuous composite beams considering slip and creep. The results suggest the following: the analytical solution is consistent with the finite element simulation results, which verifies the correctness of the analytical solution. Considering the slip and creep effects will increase the deflection of the composite beam and the bending moment of the steel beam, reduce the bending moment of the concrete slab and have a significant impact on the structural performance of the continuous composite beam. The research results considering the coupling effect of slip and creep on continuous composite beams can provide a theoretical basis for related problems.

## 1. Introduction

The steel–concrete composite structure can not only give full play to the performances of the materials themselves (tensile and compressive properties, etc.) but also to the overall superiority of the materials after combination (such as high stiffness, light weight, outstanding ductility and seismic resistance), so it is widely used in the construction industry and in bridge engineering [1,2]. In steel–concrete composite beams, shear connectors are usually used as important components to connect concrete slabs and steel beams. When the composite beam is subjected to external loads, the shear connectors produce interface slip due to longitudinal horizontal shear. The existence of interfacial slip will cause changes in the section curvature and increase the deflection of the composite beam, and it may also reduce the flexural stiffness and load-carrying capacity of the composite beam. In addition, as the loading time increases, the concrete members will creep. The creep effect will cause the redistribution of internal forces, changes in the stress state and increases in the deformation of the composite beams, etc., and in the case of super-stationary composite beams, it will cause an additional time-dependent strain on the creep secondary internal forces. It is necessary to consider the effects of interfacial slip and creep on continuous composite beams under long-term loading when investigating their mechanical properties [3,4,5,6].

Most of the studies on slip and creep effects were carried out in simply supported composite beams, and there are few studies on continuous composite beams. Dezi et al. proposed a viscoelastic analysis method and a parametric analysis of continuous composite beams with flexible shear connectors, considering the effects of creep in composite beams [7,8]. Using the variational method, Zhou et al. derived analytical expressions for deflection and slip, considering slip effects and shear deformation, but did not consider the influence of creep effects on concrete [9]. G. Ranzi et al. proposed a numerical structural formulation to compare the deformation characteristics of combined beams with partial shear interactions under short and long-term loadings [10]. Zhou et al. proposed an approximation method based on the Dischinger method to decouple the set of differential equations for concrete creep and obtain an approximate solution to the set of differential equations while neglecting the effects of redistributed bending moments on axial strain in concrete slabs [11]. Wang et al. used the age-adjusted effective modulus method to establish a differential control equation for the interface slip and axial forces of composite beams under the action of creep [12]. Considering the influences of the long-term behaviors of shrinkage and creep, Xiang et al. provided a formula for calculating the time-varying slip of externally prestressed steel–concrete composite beams [13]. Based on the linear slip model, Ranzi et al. provided an analytical solution for composite beams suitable only for distributed loads, considering the shrinkage and creep of concrete [14].

Ban et al. studied the time-dependent behaviors of simply supported composite beams with blind plate bolts under sustained loads. The results showed that the use of blind plate bolts in composite beams is beneficial to the time-dependent response [15]. Nguyen et al. developed a finite element model of a partially shear-connected composite beam, using a time-varying analysis to simulate the interaction between concrete creep and shrinkage effects and concrete cracking, and the results showed that the interaction of cracking with creep and shrinkage effects significantly increased the deflection of the composite beam [16]. Ji et al. studied the deflection of composite beams, considering the slip effect, shear deformation and concrete shrinkage and the creep effect under a long-term load, and obtained a calculation formula for composite beam deflection via the energy method. The results show that the shrinkage and creep effects of concrete have great influences on the deflection of steel–concrete composite beams [17]. Wang et al. adopted the method of adjusting the effective modulus according to age to consider the creep effect of composite box beams, and an equilibrium equation was established based on the principle of virtual work. The stiffness matrix element was obtained via the finite element method, and the slip, torsion and time-varying effects of the composite box girder were simulated [18]. Through a comparative test of two kinds of bolts, Wu et al. studied the slip effect of the negative moment zone of a composite beam to help improve stress distribution and reduce crack width [19]. M. Valente et al. used experimental tests and FE analyses to determine the influences of different shear connection degrees between concrete slabs and steel beams on the seismic responses of composite frames. The experimental results demonstrated that the slip between the slab and the beam interface contributed to the energy dissipated by the system, and the ductility demands decreased on other parts, like the beam ends and the joints [20]. Hossein Taghipoor et al. subjected 16 reinforced concrete specimens containing Forta, Basalt and Barchip fibers to experimental tests in single and hybrid groups. The effect of fiber percentage on the impact properties was studied. The results showed that the initial strength and energy absorption in hybrid-fiber-reinforced concrete increased by 45.3 and 49.7%, respectively, compared to single-fiber-reinforced concrete [21]. Abbas Sadeghian et al. used Barchip fibers, Forta and Basalt to reinforce concrete under penetration effect loading, using the Box–Behnken method and Design-Expert 13 software to study the energy absorption and impact properties. The study evaluated the fracture surface, the adhesion of fiber to the concrete and fiber degradation modes to pave the way for the optimal utilization of these hybrid FRCs [22].

Slip and creep have been considered in most composite beam research, but their combined effects can lead to inaccurate solutions under long-term loading. Compared with simply supported composite beams, the analytical problem of a continuous composite beam under the combined actions of slip and creep is difficult to solve. At present, there is no analytical solution in the literature for the internal force and deformation of a continuous composite beams under the coupling effect of slip and creep. This analytical solution is obtained in this paper. In order to accurately and conveniently calculate the coupling effect of slip and creep on steel–concrete continuous composite beams, this paper studies the mechanical properties of continuous composite beams under long-term uniform loads and establishes a governing differential equation for an axial force function for continuous composite beams via the energy variational principle. By introducing the relevant boundary conditions, the analytical solutions for the axial force, deflection and slip of a continuous composite beam under a uniform load are solved. Finally, the correctness and applicability of the obtained formulae are verified via ANSYS examples. At the same time, the effects of three kinds of shear stiffness on the slip and deflection of continuous composite beams are calculated. It is hoped that this research can provide a theoretical basis for the practical engineering calculation of continuous composite beams under the influence of a long-term load.

## 2. Establishing Control Differential Equations

In this section, based on the elastic theory and the age-adjusted effective modulus method, the basic assumptions of continuous composite beams are established, and the related theories of the slip and creep of continuous composite beams are described. Using the energy variational principle, the strain energy equation of the composite beam considering the effects of slip and creep is established, and the control differential equation and boundary conditions are obtained via the variational method and the partial integral method.

### 2.1. Basic Assumptions

In the coupled analysis of the slip and creep effects of continuous steel–concrete composite beams, the calculation is based on the following basic assumptions [23,24]:Without considering the influence of transverse shear deformation in the composite beam, the curvatures of the steel beam and the concrete flange plate are completely consistent, and the vertical lifting phenomenon at the interface of the steel–concrete composite beam is ignored.The cross-sections of the concrete slab and the steel beam meet the plane section assumption, respectively, and the stud connector conforms to the elastic sandwich setting.The stress–strain relationship between the steel beam and the concrete is linear throughout the stressing phase, and the concrete is not cracked or spalled.The shear connectors are evenly distributed along the length direction of the composite beam, and the shear force of each shear connector is linearly related to the slip.

### 2.2. Coupling Analysis Considering the Effects of Slip and Creep 

The section and element force diagram of the composite beam is shown in Figure 1. The composite beam with an elastic shear connection produces bending deformation and interface slip under a load. In this section, the axial force is selected as the unknown function according to the internal force balance condition of the composite beam, and the differential control equation for solving the unknown quantity is established via the energy variational principle [25]. In the figure, $h_{c}$ and $h_{s}$ are the heights of the concrete slab and steel beam, respectively; $b_{c}$ and $b_{f}$ are the widths of the concrete slab and steel beam respectively; N, M and *V* are the axial force, bending moment and shear force of the composite beam; $k_{s}$ is the shear stiffness of the shear connector and μ is the axial horizontal displacement of the composite beam. d is the distance between the center of gravity of the steel beam and the concrete slab, and $d=(h_{c}+h_{s})/2$.

From the diagram of the forces on the unit and the equilibrium relationship, it is known that:(1)$$N_{c}=-N_{s}=N$$

From the bending moment equilibrium condition, ∑M=0, it is known that:(2)$$M=M_{s}+M_{c}+N\cdot d$$

In Equation (2), $N_{c}=E_{c}A_{c}\varepsilon_{c}$,$N_{s}=E_{s}A_{s}\varepsilon_{s}$,$M_{c}=\varphi E_{c}I_{c}$ and $M_{s}=\varphi E_{s}I_{s}$. Here, φ denotes the curvature of the composite beams; $\varepsilon_{c}$ and $\varepsilon_{s}$ are the axial strains of the concrete slab and steel beam respectively; $E_{c}A_{c}$ and $E_{s}A_{s}$ are the axial stiffnesses of the concrete slab and steel beam; $E_{c}I_{c}$ is the bending stiffness of the concrete flange plate and $E_{s}I_{s}$ is the bending stiffness of the steel beam.

The shear connector of the composite beam is similar to the assumption that the spring conforms to the elastic interlayer. The displacement diagram of the interlayer slip is shown in Figure 2. In the figure, S is the interface slip of the composite beam, θ is the displacement rotation angle of the composite beam and $\mu_{s}$ and $\mu_{c}$ are the horizontal displacements of the steel beam and concrete slab, respectively [3].

According to the relationship between the axial strain and the displacement of the composite beam, ${\mu^{\prime }}_{c}=\varepsilon_{c}$ and ${\mu^{\prime }}_{s}=\varepsilon_{s}$, combined with the relationship shown in the figure, we can obtain:(3)$${\bar{\mu }}_{s}=\mu_{s}-(h_{s}/2)\theta$$
(4)$${\bar{\mu }}_{c}=\mu_{c}-(h_{c}/2)\theta$$
(5)$$S={\bar{\mu }}_{s}-{\bar{\mu }}_{c}=\mu_{s}-\mu_{c}+\theta d$$

On the basis of considering the influence of the interface slip of the composite beam, the age-adjusted effective modulus method (AEMM) is used to consider the creep effect of the composite beam concrete. This method uses the integral mean value theorem to transform the integral equation into an algebraic equation. At the same time, the influence of the aging properties of the concrete is considered, which simplifies the calculation and meets the needs of calculation accuracy. It is suitable for finite element analysis. The algebraic constitutive relation is [26,27]:(6)$$\varepsilon_{c}(t)=\frac{\sigma_{c}(t_{0})}{E_{c}}[1+\varphi (t,t_{0})]+\frac{\sigma_{c}(t)-\sigma_{c}(t_{0})}{E_{c}}[1+\rho (t,t_{0})\varphi (t,t_{0})]$$
where $t_{0}$ is the concrete age at loading (d), t is the calculated age of the concrete at the moment of consideration (d), $\sigma_{c}(t_{0})$ is the concrete stress at $t_{0}$, $\sigma_{c}(t)$ is the concrete stress at t, $\varphi (t,t_{0})$ represents the creep coefficient value in the whole creep process, $E_{c}$ is the elastic modulus of the concrete at the initial loading time and $\rho (t,t_{0})$ is the aging coefficient.

Through the effective modulus of age adjustment, considering the influence of the aging coefficient on the creep coefficient, we can rewrite Equation (6) as:(7)$$\varepsilon_{c}(t)=\frac{\sigma_{c}(t_{0})}{E_{c}}[1+\varphi (t,t_{0})]+\frac{\Delta \sigma_{c}(t,t_{0})}{E_{\varphi }(t,t_{0})}$$
(8)$$E_{\varphi }(t,t_{0})=\frac{E_{c}(t_{0})}{1+\rho (t,t_{0})\varphi (t,t_{0})}$$
where $\Delta \sigma_{c}(t,t_{0})$ is the stress increase of the concrete from time $t_{0}$ to time *t*; $E_{\varphi }(t,t_{0})$ is an age-adjusted effective modulus.

The range of the aging coefficient $\rho (t,t_{0})$ is generally considered to be 0.5~1.0 H. Torst [28] recommended that the average aging coefficient should be around 0.82; Wang et al. [29] combined the AEMM method with the subsequent flow theory to obtain an aging coefficient of 0.8. China’s “Code for Design of Highway Steel Structure Design Bridges” [30] considers the creep factor p, which is determined according to the load type, and takes a value of 1.1 when the permanent load is considered. In order to consider the effects of shrinkage and creep based on the energy variational method, an aging coefficient $\rho (t,t_{0})=1$ is taken.

The standard loading age of concrete is 7~14 days, and most concrete creep is completed within 1~2 years. The creep coefficient curves of concrete are different under different initial loading times. When the loading time of concrete is the same, the earlier the initial loading age, the greater the creep coefficient of the concrete will be [31]. The calculation of the creep coefficient is based on the expression of creep coefficient specified in “Specifications for Design of Highway Reinforced Concrete and Prestressed Concrete Bridges and Culverts (JTG3362-2018)” [32], that is:(9)$$\varphi \left(t,t_{0}\right)=\varphi_{0}\cdot \beta_{c}\left(t-t_{0}\right)$$
where $\varphi_{0}$ is the nominal creep coefficient and $\beta_{c}\left(t-t_{0}\right)$ is the coefficient of creep development with time after loading.

According to the principle of minimum potential energy, the composite beam maintains an equilibrium state when subjected to external forces. Therefore, this paper will establish the strain energy equations of the flange and steel beam of the steel–concrete composite beam.

The strain energy of the concrete slab:(10)$$\Pi_{c}=\frac{1}{2}\int_{0}^{L}E_{\varphi }A_{c}{\mu_{c}^{\prime }}^{2}dx+\frac{1}{2}\int_{0}^{L}E_{\varphi }I_{c}{w^{\prime \prime }}^{2}dx$$

The strain energy of steel beam:(11)$$\Pi_{s}=\frac{1}{2}\int_{0}^{L}E_{s}A_{s}{\mu_{s}^{\prime }}^{2}dx+\frac{1}{2}\int_{0}^{L}E_{s}I_{s}{w^{\prime \prime }}^{2}dx$$

The elastic interlayer slip’s strain energy:(12)$$\Pi_{sc}=\frac{1}{2}\int_{0}^{L}k_{s}{\left[(\mu_{s}-\mu_{c})+dw^{\prime }\right]}^{2}dx$$

The external load potential energy:(13)$$\Pi_{p}=-\int_{0}^{L}M(x)w^{\prime \prime }dx$$

The total potential energy of the combined beam is $\Pi =\Pi_{c}+\Pi_{s}+\Pi_{cs}+\Pi_{p}$,
(14)$$\begin{array}{l} \Pi =\frac{1}{2}\int_{0}^{L}E_{\varphi }A_{c}{\mu_{c}^{\prime }}^{2}dx+\frac{1}{2}\int_{0}^{L}E_{\varphi }I_{c}{w^{\prime \prime }}^{2}dx+\frac{1}{2}\int_{0}^{L}E_{s}A_{s}{\mu_{s}^{\prime }}^{2}dx+\frac{1}{2}\int_{0}^{L}E_{s}I_{s}{w^{\prime \prime }}^{2}dx\\ \;+\frac{1}{2}\int_{0}^{L}k_{s}{\left[(\mu_{s}-\mu_{c})+dw^{\prime }\right]}^{2}dx-\int_{0}^{L}M(x)w^{\prime \prime }dx \end{array}$$
where $E_{s}$ is the elastic modulus of the steel beam, $I_{s}$ and $I_{c}$ are the moments of inertia of the steel beam and concrete slab, respectively, $A_{c}$ and $A_{s}$ are the cross-sectional areas of the steel beam and concrete slab, respectively. $k_{s}$ is the unit beam length slip stiffness, $k_{s}=K/e$, *K* is the single shear connector connection stiffness and *e* is the shear connector spacing.

According to the principle of minimum potential energy, the first-order variation of the total potential energy is divided into zero, that is, δΠ=0. The following formula is obtained via the variational method and the integral calculation of Equation (14):(15)$$\begin{array}{l} \delta \Pi =E_{s}A_{s}\mu_{s}^{\prime }\delta \mu_{s}|{}_{0}^{L}+E_{\varphi }A_{c}\mu_{c}^{\prime }\delta \mu_{c}|{}_{0}^{L}+(E_{\varphi }I_{c}w^{\prime \prime }+E_{s}I_{s}w^{\prime \prime }-M(x)w^{\prime \prime })\delta w^{\prime }|{}_{0}^{L}\\ \;+(E_{\varphi }I_{c}w^{\prime \prime \prime }-E_{s}I_{s}w^{\prime \prime \prime }+k_{s}d\left[(\mu_{s}-\mu_{c})+dw^{\prime }\right]+M^{\prime }(x)w^{\prime \prime \prime })\delta w|{}_{0}^{L}\\ \;+\int_{0}^{L}\left[-E_{s}A_{s}\mu_{s}^{\prime \prime }+k_{s}\left[(\mu_{s}-\mu_{c})+dw^{\prime }\right]\right]\delta \mu_{s}dx\\ \;+\int_{0}^{L}\left[-E_{\varphi }A_{c}\mu_{c}^{\prime \prime }-k_{s}\left[(\mu_{s}-\mu_{c})+dw^{\prime }\right]\right]\delta \mu_{c}dx\\ \;+\int_{0}^{L}\left[E_{\varphi }I_{c}w^{\left(4\right)}+E_{s}I_{s}w^{\left(4\right)}-k_{s}d\left[(\mu_{s}-\mu_{c})+dw^{\prime \prime }\right]-M^{\prime \prime }(x)w^{\left(4\right)}\right]\delta wdx=0 \end{array}$$

The governing differential equations and boundary conditions can be obtained by Equation (15):(16)$$\{\begin{array}{c} \begin{array}{l} -E_{s}A_{s}\mu_{s}^{\prime \prime }+k_{s}\left[(\mu_{s}-\mu_{c})+dw^{\prime }\right]=0\\ -E_{\varphi }A_{c}\mu_{c}^{\prime \prime }-k_{s}\left[(\mu_{s}-\mu_{c})+dw^{\prime }\right]=0\\ EIw^{\left(4\right)}-k_{s}d\left[(\mu_{s}^{\prime }-\mu_{c}^{\prime })+dw^{\prime \prime }\right]-M^{\prime \prime }(x)w^{\left(4\right)}=0\\ E_{s}A_{s}\mu_{s}^{\prime }\delta \mu_{s}|{}_{0}^{L}=0\\ E_{\varphi }A_{c}\mu_{c}^{\prime }\delta \mu_{c}|{}_{0}^{L}=0\\ (EIw^{\prime \prime }-M(x)w^{\prime \prime })\delta w^{\prime }|{}_{0}^{L}=0\\ \left[-EIw^{\prime \prime \prime }+k_{s}d\left[(\mu_{s}-\mu_{c})+dw^{\prime }\right]+M^{\prime }(x)w^{\prime \prime \prime }\right]\delta w|{}_{0}^{L}=0 \end{array} \end{array}$$
where $EI=E_{\varphi }I_{c}+E_{s}I_{s}$. After the first derivative of the first term in Equation (16), we obtain:(17)$$-E_{s}A_{s}\mu_{s}^{\prime \prime \prime }+k_{s}\left[(\mu_{s}^{\prime }-\mu_{c}^{\prime })+dw^{\prime \prime }\right]=0$$

According to the internal force balance relationship of the composite beam, $N_{c}=-N_{s}=N$. Among them, $N_{c}=E_{\varphi }A_{c}{\mu^{\prime }}_{c}$ and $N_{s}=E_{s}A_{s}{\mu^{\prime }}_{s}$. Equation (17) can be simplified to obtain:(18)$$N^{\prime \prime }-\frac{1}{EA}k_{s}N+k_{s}dw^{\prime \prime }=0$$

In Equation (18), $\frac{1}{EA}=\frac{1}{E_{\varphi }A_{c}}+\frac{1}{E_{s}A_{s}}$.

According to the relationship between the bending moment and curvature:(19)$$EIw^{\prime \prime }+Nd=M$$

Substituting Equation (19) into Equation (18), we can obtain:(20)$$N^{\prime \prime }(x)-k_{s}(\frac{1}{EA}+\frac{d^{2}}{EI})N(x)+\frac{k_{s}d}{EI}M(x)=0$$

Simplifying Equation (20) by $\omega^{2}=\frac{k_{s}}{EA}+\frac{k_{s}d^{2}}{EI}$,$\gamma =\frac{EAd}{EI+EAd^{2}}$, we can obtain a control differential equation with the axial force N(x) as an unknown function:(21)$$N^{\prime \prime }(x)-\omega^{2}N(x)+\omega^{2}\gamma M(x)=0$$

$\omega^{2}$ is the characteristic coefficient, which is equivalent to the ratio of the relative axial stiffness to the relative bending stiffness. In this paper, the axial force of the rigid shear connection can be obtained via the product of γ, which is expressed by the differentiation γM(x).

After simplifying the first derivative of Equation (16), we obtain:(22)$$S^{\prime }(x)=-\frac{N^{\prime \prime }(x)}{k_{s}}$$

After substituting Equation (20) into Equation (22) and calculating the first derivative, we obtain:(23)$$S^{\prime \prime }(x)=-(\frac{1}{EA}+\frac{d^{2}}{EI})N^{\prime }(x)+\frac{d}{EI}M^{\prime }(x)$$

Substituting the first term of Equation (16) into Equation (23), the slip differential equation is obtained:(24)$$S^{\prime \prime }(x)-(\frac{k_{s}}{EA}+\frac{k_{s}d^{2}}{EI})S(x)=\frac{d}{EI}M^{\prime }(x)$$

Simplifying Equation (20) by $\alpha =\frac{d}{EI}$, we can obtain:(25)$$S^{\prime \prime }(x)-\omega^{2}S(x)=\alpha M^{\prime }(x)$$

In the case of determining the structural form and load, arrange the integral using the boundary conditions and axial force control differential equations to obtain the expression of the deflection function:(26)$$w(x)=\iint \frac{M(x)-d\cdot N(x)}{EI}dxdx$$

## 3. Analytical Solution of Continuous Composite Beams under a Uniform Load

In this section, according to the differential equations of the axial force, slip and deflection solved above, an analytical calculation of continuous composite beams under the combined actions of slip and creep is carried out. By substituting different boundary conditions, the analytical solution for a continuous composite beam under a uniform load is obtained.

A two-span continuous beam subjected to uniform load q is selected as the research object, as shown in Figure 3. The bearing reaction forces of the continuous composite beams A, B and C are obtained via the force method and are $R_{A}$=$R_{C}$=$\frac{3qL}{8}$ and $R_{B}$=$\frac{5qL}{4}$, respectively.

The bending moment function of the continuous composite beam is shown in Equation (27):(27)$$M=\{\begin{array}{l} M_{1}=R_{A}x-\frac{qx^{2}}{2}\;(0\leq x\leq L)\\ M_{2}=R_{A}x+R_{B}(x-L)-\frac{qx^{2}}{2}\;(L\leq x\leq 2L) \end{array}$$

### 3.1. Analytical Solution of Axial Force

The bending moment function (27) of the composite beam is substituted into the differential Equation (21) of the axial force function. Then, with the boundary conditions x=0 and $N_{1}(0)=0$, when x=L and $N_{1}(L)=N_{2}(L)$ and ${N^{\prime }}_{1}(L)={N^{\prime }}_{2}(L)$ when x=2L and $N_{2}(2L)=0$, the general solution of the axial force can be obtained:(28)$$N_{1}=\frac{\gamma q\cosh(\omega (L-x))}{\omega^{2}\cosh(L\omega )}-\frac{\gamma \sinh(\omega x)(R_{A}-Lq)}{\omega \cosh(L\omega )}+\frac{\gamma ((R_{A}x-\frac{qx^{2}}{2})\omega^{2}-q)}{\omega^{2}}$$
(29)$$\begin{array}{l} N_{2}=\frac{\gamma q\cosh(\omega (L-x))}{\omega^{2}\cosh(L\omega )}+\frac{\gamma ((R_{A}x+R_{B}(x-L)-\frac{qx^{2}}{2})\omega^{2}-q)}{\omega^{2}}\\ \;-\frac{\gamma (\frac{1}{2}L\omega (R_{A}+\frac{1}{2}R_{B}-Lq)\sinh(\omega x)-R_{B}\sinh(L\omega )\sinh(\omega (2L-x)))}{\omega \sinh(2L\omega )} \end{array}$$

### 3.2. Analytical Solution of Slip

The first derivative of the bending moment function (27) of the composite beam is substituted into the differential equation of the slip function (21). Then, through the boundary conditions x=0 and ${S^{\prime }}_{1}(0)=0$, when x=L and $S_{1}(L)=S_{2}(L)=0$ and when x=2L and ${S^{\prime }}_{2}(2L)=0$, the general solution of the slip can be obtained:(30)$$S_{1}=\frac{\alpha q\sinh(\omega (L-x))}{\omega^{3}\cosh(L\omega )}+\frac{\alpha (R_{A}-qL)\cosh(\omega x)}{\omega^{2}\cosh(L\omega )}-\frac{\alpha (R_{A}-qx)}{\omega^{2}}$$
(31)$$S_{2}=\frac{\alpha q\sinh(\omega (L-x))}{\omega^{3}\cosh(L\omega )}+\frac{\alpha (R_{A}+R_{B}-qL)\cosh(\omega (2L-x))}{\omega^{2}\cosh(L\omega )}-\frac{\alpha (R_{A}+R_{B}-qx)}{\omega^{2}}$$

### 3.3. Analytical Solution of Deflection

According to the deflection function expression in Equation (32), the bending moment function in Equation (33) under the uniform load of the composite beam and the analytical solution of the axial force, shown in Equations (28) and (29), are introduced, and then through the boundary conditions x=0 and $w_{1}(0)=0$, when x=L and $w_{1}(L)=w_{2}(L)=0$ when x=2L and $w_{2}(2L)=0$, the general solution of the slip can be obtained:(32)$$\begin{array}{l} w_{1}=\frac{\left(d\gamma -1)(qx^{4}-4R_{A}x^{3}-(qL^{3}-4R_{A}L^{2})x\right)}{24EI}+\frac{d\gamma q(x^{2}-Lx)}{2\omega^{2}EI}\\ \;+\frac{d\gamma (Lq-R_{A})(x\sinh(L\omega )-L\sinh(\omega x))}{L\omega^{3}EI\cosh(L\omega )}\\ \;+\frac{d\gamma q(x-L\cosh(\omega (L-x))+(L-x)\cosh(L\omega ))}{L\omega^{4}EI\cosh(L\omega )} \end{array}$$
(33)$$\begin{array}{l} w_{2}=\frac{\left(d\gamma -1)(qx^{4}-4(R_{A}+R_{B})x^{3}+12R_{B}Lx^{2}+(28R_{A}-8R_{B}-15Lq)L^{2}x+(14Lq-24R_{A})L^{3}\right)}{24EI}\\ \;+\frac{d\gamma (qx^{2}+(Lq-4R_{A}-2R_{B})x-2L^{2}q+(4R_{A}+2R_{B})L)}{2\omega^{2}EI}\\ \;+\frac{d\gamma ((2(R_{A}+R_{B})+qx-2qL)\sinh(L\omega )+(Lq-R_{A}-R_{B})\sinh((2L-x)\omega )}{\omega^{3}EI\cosh(L\omega )}\\ \;-\frac{(R_{A}+R_{B})x}{L\omega^{3}EI}+\frac{d\gamma q((x-L)\cosh(L\omega )+2L-x-L\cosh(\omega (L-x)))}{L\omega^{4}EI\cosh(L\omega )} \end{array}$$

## 4. ANSYS Finite Element Analysis of Continuous Composite Beam

### 4.1. ANSYS Creep Calculation Method

There are two creep analysis methods in ANSYS: explicit and implicit time integration methods. At the same time, the equation of the creep strain rate, ${\dot{\varepsilon }}_{cr}$, is provided in the creep Equations. The APDL command or GUI can be used to set creep. TB, CREEP and TBOPT define the constitutive creep model. In this paper, the creep implicit equation calculation formula when TBOPT = 6 was adopted in which the constitutive relation of the creep effect is:(34)$${\dot{\varepsilon }}_{\mathrm{cr}}=C_{1}\sigma^{C_{2}}t^{C_{3}+1}e^{-C_{4}/T}/\left(C_{3}+1\right)$$

After the first derivative of the variable *t* in Equation (34), we obtain:(35)$$\frac{d{\dot{\varepsilon }}_{\mathrm{cr}}(t)}{dt}=C_{1}\sigma^{C_{2}}t^{C_{3}}e^{-C_{4}/T}$$

Substituting the relationship $\frac{d\varepsilon_{cr}(t)}{dt}=\frac{\sigma }{E_{c}}\cdot \frac{d\varphi (t)}{dt}$ of the stress and strain of the concrete creep coefficient in the time increment dt into Equation (35):(36)$$\frac{\sigma }{E_{c}}\cdot \frac{d\varphi (t)}{dt}=C_{1}\sigma^{C_{2}}t^{C_{3}}e^{-C_{4}/T}$$

Creep should have a linear relationship with the strain rate of concrete. Take $C_{2}=1$ and $C_{3}=0$ and assume that the creep strain rate only depends on the strain in the material, which conforms to the strain strengthening criterion. The effect of temperature on creep is not considered in the calculation of concrete. Let $C_{4}=0$, Equation (36):(37)$$C_{1}=\frac{d\varphi (t)}{dt}\cdot \frac{1}{E_{c}}=\frac{\varphi_{n-1}-\varphi_{n}}{t_{n-1}-t_{n}}\cdot \frac{1}{E_{c}}$$

According to the expression, we can see the relationship between $C_{1}$, the variable time *t* and the creep coefficient. The time at the *i*th moment is $t_{i}$, and the creep coefficient is $\varphi_{i}$. The calculation equation of the parameter $C_{1i}$ can be expressed as:(38)$$\{\begin{array}{l} C_{11}=\frac{\varphi_{1}}{t_{1}-t_{0}}\cdot \frac{1}{E_{c}},\;\;\;\;i=1\\ C_{1i}=\frac{\varphi_{i}-\varphi_{i-1}}{t_{i}-t_{i-1}}\cdot \frac{1}{E_{c}},\;i\geq 2 \end{array}$$

For the calculation of the creep coefficient $\varphi (t,t_{0})$ in ANSYS, according to the creep coefficient calculation formula of a bridge specification (JTG3362), the relevant formula is input by writing an APDL command loop statement. The constitutive relation of the theoretical calculation is calculated by the expression $C_{1i}$. The change in the creep effect process with time is simulated by constantly changing the material properties.

### 4.2. Continuous Composite Beam Example Verification

The section size of the steel–concrete continuous composite beam selected in this paper is shown in Figure 4. The correctness of the theoretical solution formula is verified via the finite element software ANSYS 18.0. The concrete grade is C30, the elastic modulus is $E_{c}=3.0\times 10^{4}$ N/mm^2^, the Poisson’s ratio value is $\mu_{c}=0.2$, the elastic modulus of the steel beam is $E_{s}=2.1\times 10^{5}$ N/mm^2^ and the Poisson’s ratio value is $\mu_{s}=0.3$. The shear connectors are made of 19 mm diameter pegs which are arranged evenly in a single row along the length of the beam with a spacing length of e=200mm. The shear stiffness is taken as $K=5.27\times 10^{4}$ N/mm.

A sketch of the continuous composite beam model under a uniform load is shown in Figure 5. The total length of the continuous composite beam is 16 m, the single span length is 8 m, and the uniform load is q = 50 KN/m. The average annual ambient humidity is RH = 50%, the loading age $t_{0}$=14 days and dt=14 days, calculated as 728 days. The creep coefficient reflects the change in concrete with time under a long-term load, and the creep coefficient increases obviously with the continuous action of the load during the loading age. The value of the creep coefficient calculated in the example is 2.8 at 728 days, and the change in the creep coefficient during the loading time is shown in Figure 6.

The ANSYS finite element composite beam model was built for structural analysis. The upper and lower flanges of both the concrete slab and the steel beam were simulated using the solid unit Solid45, and the web of the steel beam was simulated using the Plane42 unit; the studs between the concrete slab and the steel beam were simulated using the Combin39 spring element. Combin39 can define the load–slip curve with an axial or torsional function and can provide good simulations of studs in composite beams. By controlling the meshing and nodes of the steel beam and the concrete slab, the steel beam and the concrete node (two overlapping nodes) were selected at the position at which the studs were arranged, and the spring element Combine39 element was established. The spring element node, the concrete node and the steel beam node were connected by the degree of freedom coupling. The constraint arrangement was a fixed hinge support at the left end of the continuous beam and movable hinge supports at the mid-span and right end. The finite element model of the continuous composite beam is shown in Figure 7.

The results of the comparison between the theoretical and FEM calculated axial forces at the mid-span position of the left span of the continuous composite beam are shown in Table 1 for day 14 and day 728 under uniform loads. The axial force curve with time variation at the mid-span position of the left span is shown in Figure 8.

As can be seen from Table 1, the error between the axial force values calculated from the theoretical equations and the finite element calculations is within 2%, which shows a good agreement and verifies the correctness of the analytical solution of the axial force.

As can be seen in Figure 8, the axial force curves obtained from the analytical solution and the ANSYS calculations follow the same trend. Under the coupled effects of creep and slip, the axial force in the theoretical calculation of the continuous composite beam increases by 1.72%, while the value of the axial force in the ANSYS simulation increases by 0.38%. This indicates that the axial force of the continuous composite beam is increased by creep, but the impact is small.

The comparative results of deflection and slip at the mid-span position of the left span of the continuous composite beam are shown in Table 2, and the deflection–time and slip–time curves of the composite beam are shown in Figure 9.

As can be seen from Table 2, the error in the deflection values between the analytical solution and the finite element calculations is within 8%, and the error in the slip values is around 6%. This shows that the analytical solution and the finite element simulations are in good agreement and satisfy the need for correctness and calculation accuracy.

As can be seen in Figure 9, the deflection and slip curves for the analytical and finite element calculations follow the same trend. In Figure 9a, the deflection values for the theoretical calculations increased by 18.43% and the deflection values for the finite element calculations increased by 14.36% from the initial moment of creep to after 728 days of creep, indicating that the deflection of the continuous composite beam continued to increase under the action of creep. In Figure 9b, it can be seen that the slip growth is slow, with 1.27% growth in the theoretically calculated slip and 1.96% growth in the finite element calculated slip over the 728 days of loading time. This shows that the effect of creep on slip is small.

### 4.3. Example Analysis of Continuous Composite Beam

The shear stiffness of the shear connector was taken as the research variable, and the remaining parameters were the same as in the previous example. Three kinds of shear stiffness, namely, $K_{1}=2.47\times 10^{4}$ N/mm, $K_{2}=5.27\times 10^{4}$ N/mm and $K_{3}=1.0\times 10^{5}$ N/mm, were selected. The changes in the slip, deflection and axial force of the continuous composite beams were observed at the initial time of creep (14 d) and the final loading time (728 d) under different shear stiffness, and the bending moment changes in the composite beams were analyzed.

The maximum axial forces of the composite beam at the initial time and the final time of creep under the three shear stiffnesses are compared, as shown in Table 3. The distribution of the axial force of the composite beam along the length of the beam is shown in Figure 10. The dotted line in the figure is the axial force at 14 d, at the initial time of creep, and the solid line is the axial force at 728 d, at the final time of creep loading age.

According to the data in Table 3 and the axial force distribution in Figure 10, it can be seen that the maximum axial force of the continuous composite beam is at the beam end near the mid-span of the half-span. The axial force of the composite beam increases with the increase in shear stiffness, and the influence of the rate of the effect of creep on the axial force of the composite beam decreases with the increase in shear stiffness, which decreases by about 3%.

In the case of different shear stiffnesses, the maximum deflection and slip values of the composite beam on the 14th and 728th days are compared, as shown in Table 4. The distribution of slip and deflection along the length of the beam is shown in Figure 11. The dotted line in the figure is the initial time of creep, 14 d, and the solid line is the final time of creep loading age, 728 d.

According to Table 4 and Figure 11a, it can be seen that the slip of the composite beam decreases with an increase in the shear stiffness, and the larger slip values are at the end bearing of the continuous composite beam and the position near the mid-span bearing. Increasing the shear stiffness can reduce the influence of creep on slip. In the case of small shear stiffness, considering the creep effect will increase the slip of the composite beam; when the shear stiffness is large, the creep effect has little effect on the slip and can reduce the slip.

According to Table 4 and Figure 11b, it can be seen that the maximum deflection of the continuous composite beam occurs near the middle of the half-span. The creep will increase the deflection of the composite beam, and the increase in the shear stiffness will reduce the deflection of the composite beam, and this degree of reduction is about 12%. The influence of the rate of creep on the deflection of a composite beam will not decrease with the increase in shear stiffness but will increase.

A comparison between the concrete bending moment of the composite beam and the bending moment of the steel beam at the initial moment and at the final moment of creep is shown in Table 5. The concrete bending moment and the steel beam bending moment along the length of the beam are shown in Figure 12. 

According to Table 5 and Figure 12, it can be seen that the maximum bending moment of the continuous composite beam occurs at the mid-span support. The increase in shear stiffness will reduce the bending moment of the concrete and the bending moment of the steel beam, and this reduction degree is about 9%. The creep will reduce the bending moment of the concrete and increase the bending moment of the steel beam. The influence of the rate of creep on the concrete bending moment is about 49%, and the influence of the rate on the steel beam bending moment is about 12%, which is less affected by the change in shear stiffness.

## 5. Conclusions

In this paper, the slip, deflection and axial force of steel–concrete continuous composite beams under the coupling of slip and creep effects are analyzed and studied. The conclusions are as follows:In this paper, the energy variational method is used to establish the control differential equation, considering the combined actions of slip and creep. The theoretical analytical solutions for the slip, deflection and axial forces of continuous composite beams under uniform loads are derived. The correctness of the analytical solution is verified by establishing a finite element composite beam model.For a composite beam under a long-term load, creep will increase the deflection of the composite beam and the steel beam bending moment, reducing the concrete bending moment. The deflection of the composite beam increases by 14% under the influence of creep, the bending moment of the steel beam increases by 12% and the bending moment of the concrete decreases by 49%. Creep has little effect on the slip and axial force of a composite beam, increasing by about 2%.The analytical solution obtained in this paper is suitable for the calculation and analysis of any equal-span continuous composite beam under a uniform load and can calculate any shear stiffness, which can quickly and intuitively reflect changes in the deflection, slip and axial force under a long-term load. The increase in the shear stiffnesses of shear connectors will increase the axial forces of composite beams and reduce slip, deflection and bending moments. The increase in shear stiffness will reduce the influence of creep on the axial forces and slip of composite beams and increase the influence of creep on deflection.

In this paper, mechanical properties under the coupling actions of slip and creep are studied. Future work will carry out experimental research on continuous composite beams. The changes in the internal forces in the negative moment zones of composite beams under coupling and the stress caused by concrete cracking affected by creep must be further studied.

## Figures and Tables

**Figure 1 materials-16-04741-f001:**
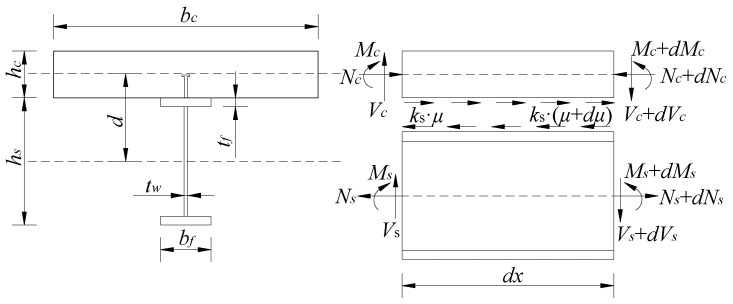
Schematic diagram of the combined beam cross-section and microsegment subjected to forces.

**Figure 2 materials-16-04741-f002:**
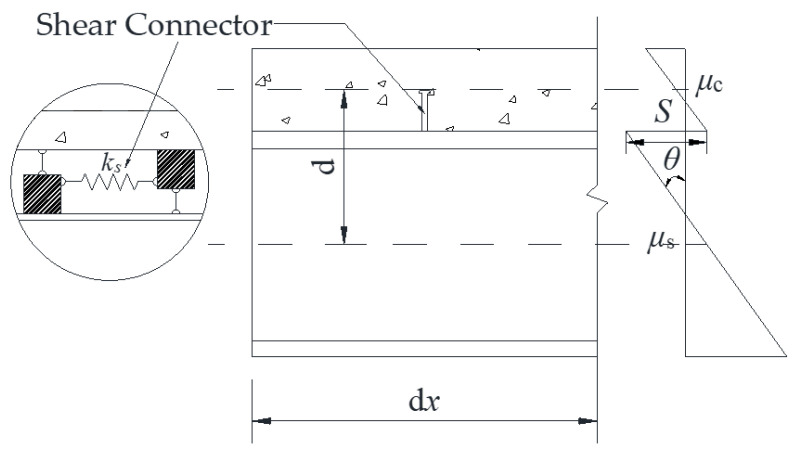
Shear connection and longitudinal displacement diagram of composite beam.

**Figure 3 materials-16-04741-f003:**
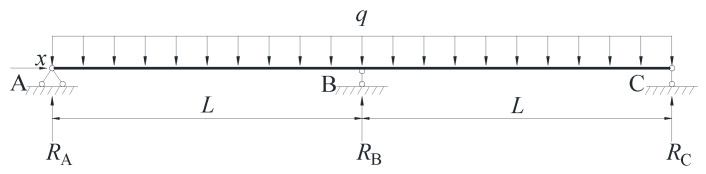
Two-span continuous composite beam under uniform load.

**Figure 4 materials-16-04741-f004:**
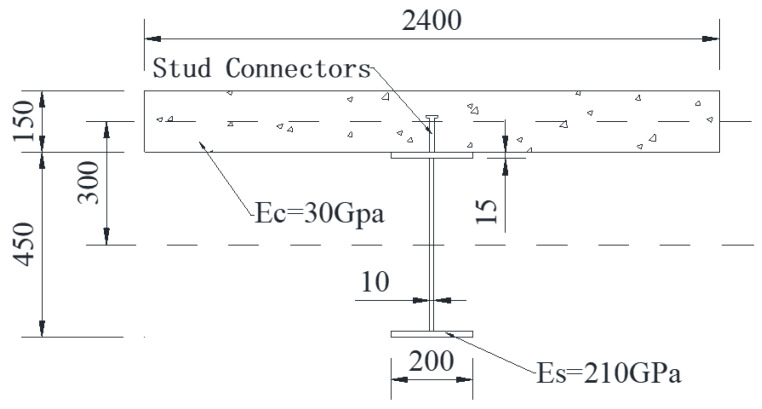
Section dimensions of the continuous composite beam (unit: mm).

**Figure 5 materials-16-04741-f005:**
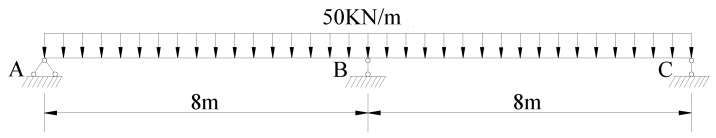
Section dimensions of the continuous composite beam.

**Figure 6 materials-16-04741-f006:**
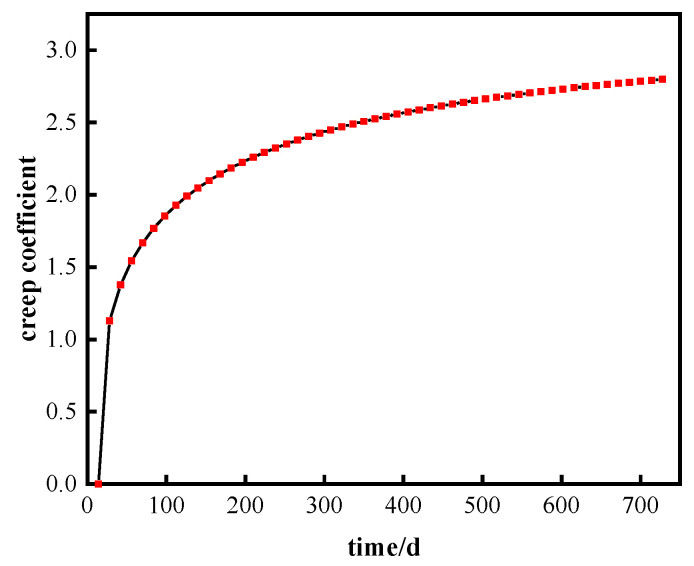
Change in creep coefficient over the loading time.

**Figure 7 materials-16-04741-f007:**
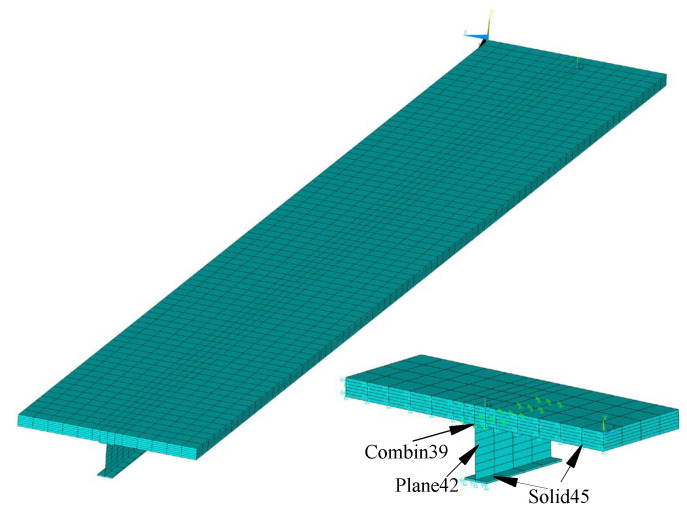
Finite element model of a continuous steel–concrete composite beam.

**Figure 8 materials-16-04741-f008:**
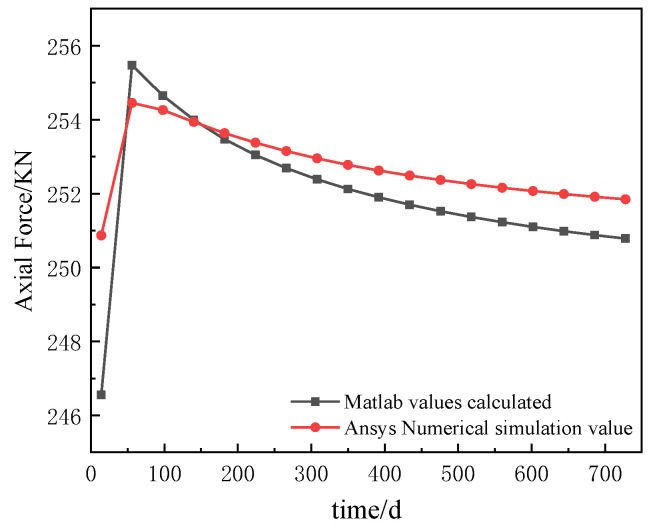
Comparison of axial force of left span of continuous composite beam.

**Figure 9 materials-16-04741-f009:**
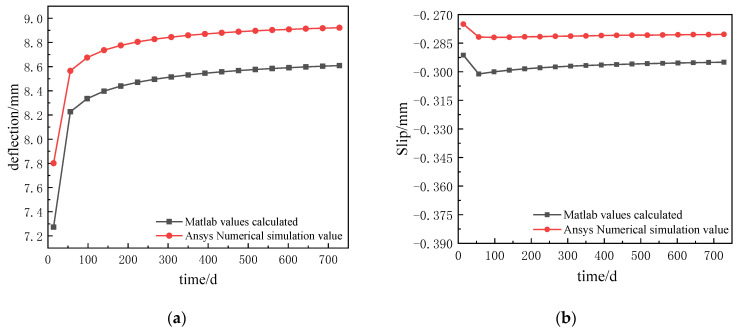
Comparison of mid-span deflection and slip in the left span of the continuous composite beam: (**a**) comparison of deflection–time curves; (**b**) comparison of slip–time curves.

**Figure 10 materials-16-04741-f010:**
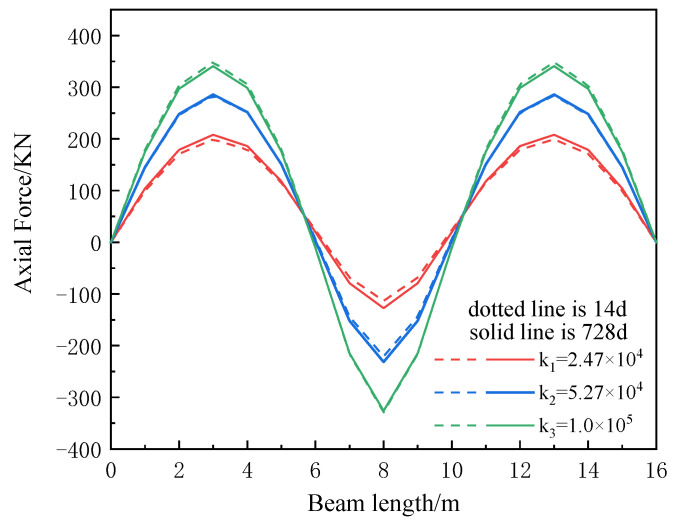
Axial force distribution of composite beams under different shear stiffnesses.

**Figure 11 materials-16-04741-f011:**
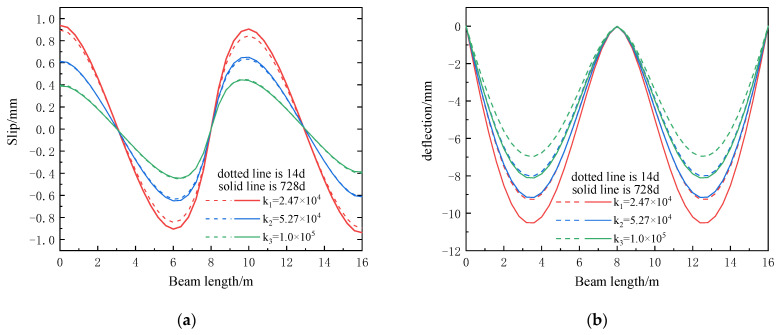
Slip and deflection distribution of composite beams under different shear stiffnesses: (**a**) slip distribution diagram; (**b**) deflection distribution diagram.

**Figure 12 materials-16-04741-f012:**
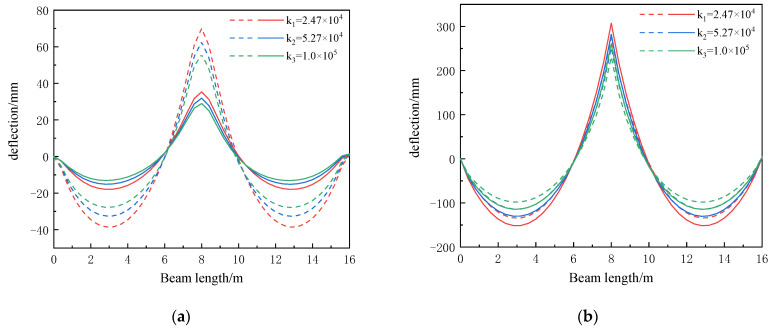
Moment distribution of composite beams under different shear stiffnesses: (**a**) concrete bending moment distribution diagram; (**b**) steel beam bending moment distribution diagram.

**Table 1 materials-16-04741-t001:** Comparison of axial forces under uniformly distributed loads.

Time/d	$\mathrm{Simulated}\;\mathrm{Calculation}\;\mathrm{Value}\;z_{1}/\mathrm{KN}$	$\mathrm{ANSYS}\;\mathrm{Values}\;\mathrm{Calculation}\;z_{2}/\mathrm{KN}$	Error
14	246.55	250.87	1.75%
728	250.78	251.84	0.42%

Notes: Error = $\left|(z_{2}-z_{1})/z_{1}\right|$.

**Table 2 materials-16-04741-t002:** Comparison of deflection and slip under uniformly distributed loads (mm).

	Simulated Calculation Value		ANSYS Values Calculation		Error	
Time/d	$\mathrm{Deflection}\;z_{1}$	$\mathrm{Slip}\;z_{2}$	$\mathrm{Deflection}\;z_{3}$	$\mathrm{Slip}\;z_{4}$	Error 1	Error 2
14	7.27	−0.2913	7.80	−0.275	7.29%	5.60%
728	8.61	−0.2950	8.92	−0.2804	3.6%	4.95%

Notes: Error 1 = $\left|(z_{3}-z_{1})/z_{1}\right|$; Error 2 = $\left|(z_{4}-z_{2})/z_{2}\right|$.

**Table 3 materials-16-04741-t003:** Comparison of the maximum axial forces of composite beams under different shear stiffnesses.

	Axial Force/KN		
Time/d	$K_{1}$	$K_{2}$	$K_{3}$
14	199.20	284.10	348.25
728	208.02	285.98	340.77
Change rate	4.43%	0.67%	−2.15%

**Table 4 materials-16-04741-t004:** Comparison of maximum slip and deflection values of composite beams under different shear stiffnesses.

	Slip/mm			Deflection/mm		
Time/d	$K_{1}$	$K_{2}$	$K_{3}$	$K_{1}$	$K_{2}$	$K_{3}$
14	0.844	−0.634	−0.446	−9.27	−8.0	−6.96
728	−0.905	−0.650	−0.439	−10.51	−9.16	−8.11
Change rate	7.23%	2.52%	−1.57%	13.38%	14.50%	16.52%

**Table 5 materials-16-04741-t005:** Comparison of concrete bending moment and steel beam bending moment of composite beams under different shear stiffnesses.

	Concrete Bending Moment/KN/m			Steel Beam Bending Moment/KN/m		
Time/d	$K_{1}$	$K_{2}$	$K_{3}$	$K_{1}$	$K_{2}$	$K_{3}$
14	70	62.64	55.92	275.20	252.0	231.0
728	35.38	31.88	28.98	307.40	282.0	260.60
Change rate	−49.46%	−49.11%	−48.18%	11.70%	11.90%	12.81%

## Data Availability

Data sharing is not applicable to this article.

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
