# Peer review of "Study on the Mechanical Properties of Continuous Composite Beams under Coupled Slip and Creep"

_materials, 2023, doi:10.3390/ma16134741_

Round 1

Reviewer 1 Report

Some points of the study should be improved or better explained. A re-review of the manuscript is required.

1) Abstract can be improved, better highlighting the main novelty aspects of the study. The main novel findings of the study should be better presented.

2) Lines 88-99. At the end of Introduction, the main aims and steps of the study are summarized. However, a further explanation and discussion of the main novelty aspects and contributions of the work can be provided. In particular, it is suggested to highlight the main advances of this study with respect to previous papers of the authors on the same topic.

3) Considering the topic addressed in this study, in Introduction it is suggested to insert a remark about the importance of composite action between concrete slab and steel beam in composite structures under seismic actions. To this aim, the following reference, which investigates, through experimental tests and FE analyses, the influence of different shear connection degrees between concrete slab and steel beam on the seismic response of composite frames, can be mentioned:

https://doi.org/10.1016/j.jcsr.2009.05.001

4) -At the beginning of Sections 2 (before “basic assumptions”), it is suggested to shortly summarize the aim and the main contribution of the section.

-At the beginning of Sections 3 (Line 248), it is suggested to shortly summarize the aim and the main contribution of the section.

5) Improve the caption of Fig. 1 and Fig. 2.

6) Line 313. Check the value of Poisson ratio for concrete.

7) A more comprehensive explanation of the diagram shown in Fig. 6 should be provided in the text. Moreover, improve the caption of Fig. 6.

8) Line 332. Different element types are used for the flanges and web of the steel beam. Provide an explanation.

9) Line 333. It is required to provide a more comprehensive explanation of “Combin39 spring units” adopted to simulate the connection between the concrete slab and the steel beam.

10) Line 339. Delete “diagram” in the caption of Fig. 7. No diagram is present…..

11) Section 5. Conclusions should be improved, better highlighting the main novelty aspects and original contributions of the study.

12) Some short recommendations for future work should be included at the end of Conclusions.

13) An extensive revision of the text should be carried out in order to improve the quality of English language and correct some grammar mistakes and typos.

13) An extensive revision of the text should be carried out in order to improve the quality of English language and correct some grammar mistakes and typos.

Author Response

Thank you for your letter dated June 23. We thank the reviewers for the time and effort that they have put into reviewing the previous version of the manuscript. Your suggestions have enabled us to improve our work. Based on the instructions provided in your letter, we have made corrected modifications on the manuscript. Meanwhile, we uploaded the file of the revised manuscript. The following content is our point-by-point response to the comments raised by the reviewers.

Reviewer 2 Report

1.Could you explain the significance of considering the combined action of slip and creep in the analysis of steel-concrete continuous composite beams under long-term load? How do these factors affect the overall mechanical behavior of the composite beams?

2.How was the energy variational method utilized to establish the control differential equation that accounts for slip and creep effects? Could you provide more details about the formulation and assumptions involved in deriving the analytical solutions for slip, deflection, and axial force under uniform load?

3.In the conclusions, you mentioned that creep has a significant impact on the deflection, bending moment of the steel beam, and bending moment of the concrete slab. Could you explain the underlying reasons for these effects? How does creep influence the time-dependent behavior of the composite beams?

4.I would suggest the authors expand and reconsider the introduction. The failure to design structures is a recent and exciting topic, which has led to many articles published in this field in recent years. So, it would be great if the authors included more literature reviews. In addition, I would use the most recent lectures or studies in the related field to add to the references. For example:

https://doi.org/10.1155/2023/7110987

https://doi.org/10.1016/j.istruc.2022.07.030

5.You also mentioned that the shear stiffness of shear connectors plays a role in the axial force, slip, deflection, and bending moment of the composite beams. Could you elaborate on the relationship between shear stiffness and these mechanical properties? How can adjusting the shear stiffness help mitigate the effects of creep on the axial force, slip, deflection, and bending moment?

Author Response

(The authors gave the same response as above.)

Round 2

Reviewer 1 Report

The authors have addressed the concerns raised by the reviewer.

The revised manuscript is recommended for publication.

A revision of English language is required.